# Beyond checklists: Using clinic ethnography to assess the enabling environment for tuberculosis infection prevention control in South Africa

Stella Arakelyan[1,2], Hayley MacGregor[3], Anna S. Voce[4], Janet Seeley[5], Alison D. Grant[6], Karina Kielmann[1,7] *

1 Institute for Global Health and Development, Queen Margaret University Edinburgh, Queen Margaret University Way, Edinburgh, United Kingdom, 2 Advanced Care Research Centre, Centre for Population Health Sciences, Usher Institute, Edinburgh University, Edinburgh, United Kingdom, 3 The Institute of Development Studies, University of Sussex, Brighton, United Kingdom, 4 Discipline Public Health Medicine, School of Nursing and Public Health, University of KwaZulu-Natal, Durban, South Africa, 5 Department of Global Health and Development, Faculty of Public Health and Policy, London School of Hygiene and Tropical Medicine, London, United Kingdom, 6 TB Centre, London School of Hygiene and Tropical Medicine, London, United Kingdom, 7 Department of Public Health, Equity & Health Unit, Institute of Tropical Medicine Antwerp, Antwerp, Belgium

* kkielmann@itg.be

**Data Availability Statement:** The qualitative data reported on in this paper was archived in the UK Data Service's Discover catalogue with a unique

## Abstract

Sub-optimal implementation of infection prevention and control (IPC) measures for airborne infections is associated with a rise in healthcare-acquired infections. Research examining contributing factors has tended to focus on poor infrastructure or lack of health care worker compliance with recommended guidelines, with limited consideration of the working environments within which IPC measures are implemented. Our analysis of compromised tuberculosis (TB)-related IPC in South Africa used clinic ethnography to elucidate the enabling environment for TB-IPC strategies. Using an ethnographic approach, we conducted observations, semi-structured interviews, and informal conversations with healthcare staff in six primary health clinics in KwaZulu-Natal, South Africa between November 2018 and April 2019. Qualitative data and fieldnotes were analysed deductively following a framework that examined the intersections between health systems 'hardware' and 'software' issues affecting the implementation of TB-IPC. Clinic managers and front-line staff negotiate and adapt TB-IPC practices within infrastructural, resource and organisational constraints. Staff were ambivalent about the usefulness of managerial oversight measures including IPC protocols, IPC committees and IPC champions. Challenges in implementing administrative measures including triaging and screening were related to the inefficient organisation of patient flow and information, as well as inconsistent policy directives. Integration of environmental controls was hindered by limitations in the material infrastructure and behavioural norms. Personal protective measures, though available, were not consistently applied due to limited perceived risk and the lack of a collective ethos around health worker and patient safety. In one clinic, positive organisational culture enhanced staff morale and adherence to IPC measures. 'Hardware' and 'software' constraints interact to impact negatively on the capacity of

DataCite Digital Object Identifier (DOI). The link to our data is: http://doi.org/10.5255/UKDA-SN-854435.

**Funding:** The support of the Economic and Social Research Council (UK) is gratefully acknowledged (SA, HM, AV, JS, AG and KK). The Umoya omuhle study is funded by the Antimicrobial Resistance Cross Council Initiative supported by the seven research councils in partnership with other funders including support from the GCRF (ref. ES/P008011/1). Additional support was received from The Bloomsbury SET (Research England; ref. CCF17-7779) for AG and KK. JS and AG receive support from Wellcome Trust Strategic Core Award (Africa Health Research Institute, ref. 201433/A/16/A). The funders had no role in study design, data collection and analysis, decision to publish, or preparation of the manuscript.

**Competing interests:** The authors have declared that no competing interests exist.

primary care staff to implement TB-IPC measures. Clinic ethnography allowed for multiple entry points to the 'problematic' of compromised TB-IPC, highlighting the importance of capturing dimensions of the 'enabling environment', currently not assessed in binary checklists.

## Introduction

In recent years, there has been heightened attention to healthcare settings as the front-line of containment and response strategies for endemic infectious diseases as well as sporadic outbreaks. Effective infection prevention and control (IPC) measures are deemed a global priority [1, 2], particularly in the light of newly emerging respiratory syndrome coronaviruses. Developing sustainable action plans to embed IPC within the structures and activities of resource-constrained health systems is critical. However, there is ample evidence to suggest that IPC guidelines for airborne diseases are challenging to put into practice [1, 3]. Poor implementation of IPC is frequently described as either a problem of inadequate resources [3–5] or a problem of behavioural non-compliance of staff to guidelines [6]. At the same time, evidence on how to create and sustain a culture of safety in resource-constrained health systems is scarce.

The World Health Organization (WHO) general guidelines on improving IPC in healthcare settings lay out a series of multi-modal strategies that comprise essential components of an IPC programme: IPC guidelines; IPC education and training; infection surveillance, monitoring and feedback [2]. For successful implementation of IPC, these programme components need to be embedded within what is referred to as an 'enabling environment'. This concept, as put forward by WHO, is limited to infrastructural and resource considerations of the health system including workload, staffing and bed occupancy, built environment and equipment. Accordingly, the implementation of IPC at the clinic level has been primarily evaluated through two predominant approaches. Commonly, structured binary checklists are used to examine, firstly, the availability or absence of specific equipment, information and human resources, and secondly, the frequency of processes intended to support the maintenance of these inputs through e.g., training, monitoring, audits, on-going surveillance [7, 8]. A less common approach uses structured tools to examine elements of the underlying organizational culture that promotes or dissuades uptake of IPC measures to reduce healthcare-associated infections (HCAI) [9, 10].

While the 'checklist' is seen as an *elegant and simple tool* to assess IPC measures [11], it reduces measures to individual components, processes or behaviours that can be viewed and evaluated in isolation, in other words, independent of the broader features and dynamics of working environments in which measures are implemented and applied. To address this gap, we suggest a systems-thinking approach that recasts IPC as a complex intervention [12] involving multiple interacting components in a real-world setting rather than a series of items on a checklist. Recent health systems frameworks offer a way of examining the dynamic inter-relationships between 'hardware' elements (human resources, medicine and technology, organisational structures, service infrastructure, information systems, financing) and 'software' elements (relationships and power, values and norms, ideas and interest) within healthcare and wider social and political contexts [13,14]. Greater attention to these dynamics offers a means to better delineate what 'enabling' (or conversely, 'disabling') environments for optimal implementation of IPC measures might comprise.

In this paper, we focus on the specific example of IPC for the prevention of *Mycobacterium tuberculosis* (*Mtb*) transmission in healthcare clinics, noted to have "*underappreciated*

*synergies*" with IPC for COVID-19) [15]. *Mtb* is a bacterial infection that spreads through inhaling tiny droplets (aerosols) from the coughs or sneezes of a person with TB. The risk of *Mtb* transmission for patients and health workers is, therefore, high in enclosed settings where people with undiagnosed active TB may congregate such as primary health clinics [16]. For TB-related IPC, the WHO (2019) recommends a hierarchy of controls that clusters measures under managerial and administrative controls, environmental controls and personal respiratory protection (Table 1).

To date, studies of TB-IPC implementation rarely consider the broader working environments and cultures within which guidelines and policies are implemented [17]. If they do, it appears difficult to move beyond recommendations that emphasise more training to ensure better 'compliance' on the part of health workers, underpinning a narrow view of IPC as primarily a behavioural rather than a systems issue. We draw on ethnographic fieldwork in South African primary care clinics, where we examined the context, processes, and practices of suboptimal implementation of tuberculosis-related IPC (TB-IPC). We specifically focused on the dynamic interactions within and across 'hardware' and 'software' elements of TB-IPC to elucidate the challenges and opportunities of sustaining the so-called 'enabling environment' for successful uptake of IPC measures in many resource-constrained settings, for TB and other airborne infections. In this work, we also demonstrate the utility of clinic ethnography to refine and extend current binary checklist approaches to assessing IPC implementation.

## Material and methods

This study forms part of an interdisciplinary, mixed-methods project entitled "Umoya omuhle: Infection Prevention and Control for Drug-Resistant Tuberculosis in South Africa in the Era of Decentralised Care: A Whole Systems Approach". We draw on the data collected between November 2018 and April 2019 during ethnographic fieldwork in primary care clinics in the Kwazulu-Natal province of South Africa.

### Study context

South Africa is a resource-constrained setting with a high TB burden; the estimated incidence of TB in 2019 was 615 per 100 000, which is among the highest in the world [18]. Healthcare-acquired *Mtb* transmission is a growing concern in the country [19]. Earlier iterations of the WHO guidelines have been adapted to the local policy context in South Africa [20]. Further, in 2015 the South African National Infection Prevention and Control Strategy for TB,

**Table 1. Hierarchy of TB-IPC controls adopted from the WHO (2019).**

| Hierarchy of TB-IPC controls | Recommended measures |
|---|---|
| Managerial and administrative controls | TB-IPC guidelines |
| | IPC champions |
| | Healthcare worker training |
| | Triage of people with TB sign and symptoms or TB disease |
| | Respiratory isolation of people with presumed or demonstrated infectious TB |
| | Prompt initiation of effective TB treatment of people with TB |
| | Respiratory hygiene |
| Environmental controls | Ultraviolet germicidal irradiation |
| | Ventilation systems |
| Personal respiratory protection | Wearing protective masks |
| | Wearing particulate respirators |

MDR-TB, XDR-TB [21] was established targeting the spread of TB and eradication of the TB epidemic by 2030. IPC is also one of the essential pillars of a major health system strengthening intervention to address current infrastructure, administrative and resource deficiencies in the quality of healthcare services, known as the Ideal Clinic Initiative (ICI) [22]. The ICI assumes reorganisation, integration of services (e.g., TB, HIV, STI, emergency, child and maternity) and improvement in safety and quality of care. In principle, these policy initiatives are intended to address long-standing structural and operational issues in healthcare settings that contribute to sub-optimal IPC and gaps in the continuity of TB care. However, the TB-IPC measures remain poorly implemented [23–25], calling for context-specific approaches to better embed IPC measures within South African clinic systems [26, 27].

KwaZulu-Natal Province (KZN) is one of the poorest provinces in South Africa [28] and one of four high-TB burden provinces, with an estimated drug-sensitive *Mtb* incidence of 525 per 100 000 population in 2017 [29]. The effective *Mtb* treatment coverage for KZN has been estimated at 56% for the 2016 to 2018 triennium [30]. Initiation and management of drug-sensitive TB have been decentralised to the clinic level, while the initiation and continuation of Multidrug-Resistant TB (MDR-TB) have been decentralised to Level 1/District Hospitals.

### Ethics statement

Ethical approvals were granted by the Biomedical Research Ethics Committee of the University of KwaZulu-Natal (REF. BE082/18); the Research Ethics Committee of Queen Margaret University (REF. REP 0233), and the Observational/Interventions Research Ethics Committee of the London School of Hygiene & Tropical Medicine (REF. 14872).

Written gatekeeper permission to conduct the study was provided at the provincial level (Ref: KZ_201810_016) and from each of the three health districts within which the six clinics were located. Informed written consent for clinic participation in the study was sought from the clinic manager of each clinic, prior to the start of fieldwork/data collection. On entry to each clinic, researchers held an information session with all staff prior to the start of data collection. This was usually incorporated into the regular staff meeting, to inform clinic staff of all researcher activities while on site. All questions pertaining to the study purpose and processes were addressed in the information session and as they arose during the fieldwork. Members of staff engaged in informal conversations could thus be informed regarding the nature of the research and their participation to ensure transparency regarding the data collection generated through the ethnographic method. Individual written informed consent was obtained prior to each in-depth interview, including permission to audio-record the interview. Assurance of anonymity and confidentiality was provided for both clinic sites and individuals within each site. Assurance was also given regarding the freedom to withdraw from the study at any time, with no ensuing negative consequence.

### Inclusivity in global research

Additional information regarding the ethical, cultural, and scientific considerations specific to inclusivity in global research is included in the (S1 Checklist).

### Data sources and collection

Six primary health clinics were selected purposively to take account of the diversity of clinics across the province. Selection was based on the year of establishment and type of facility; governance structure; daily headcount and catchment population; the presence or absence of appointment systems and the extent of the decentralisation of DR-TB services. The purposive

**Table 2. Description of clinics and data collection methods.**

| Facility | Location* (decade)** | Governance structure | Patient load*** | Data collection period | Data source | Data collection method |
|---|---|---|---|---|---|---|
| 1 | Semi-rural (1990s) | KZN government | 300 | 20–22 November 2018 | Thick description | Observation, informal conversation, fieldnotes |
| | | | | | Clinic manager (2) | Semi-structured interview |
| | | | | | Nurse | Semi-structured interview |
| | | | | | TB nurse | Semi-structured interview |
| 2 | Rural (1980s) | KZN government | 80 | 27–29 November 2018 | Thick description Clinic manager | Observation, fieldnotes |
| | | | | | | Interview |
| | | | | | TB nurse Speech therapist | Informal conversation |
| | | | | | | Informal conversation |
| 3 | Peri-urban (2000s) | KZN government | 950 | 4–6 December 2018 | Thick description Clinic manager | Observations, fieldnotes |
| | | | | | | Semi-structured interview |
| | | | | | Doctor | Informal conversation |
| | | | | | TB nurse | Informal conversation |
| | | | | | IPC manager | Informal conversation |
| | | | | | HR manager | Informal conversation |
| | | | | | Radiographer | Informal conversation |
| 4 | Urban (1980s) | KZN government | 1000 | 5–8 February 2019 | Thick description | Observation, fieldnotes |
| | | | | 7 March 2019 | Nursing service manger | Semi-structured interview |
| | | | | 3 April 2019 | IPC manager | Semi-structured interview |
| | | | | 10 April 2019 | | |
| 5 | Urban (1980s) | KZN government | 300 | 12 February 2019 | Senior health worker | Semi-structured interview |
| 6 | Rural (2000s) | KZN government | 80 | 19 February 2019 | Thick description | Observation, fieldnotes |
| | | | | | Clinic manager | Semi-structured interview |

\* The clinic location is based on the classification adopted by the National Treasury Republic of South Africa in Local Government Budgets and Expenditure Review (2011). Based on this classification, rural refers to sparsely populated areas in which people depend on farming, natural resources and migratory labour, remittances and social grants; semi-rural refers to areas that are functionally rural and have traditional tenure systems; urban refers to large towns with an urban core; peri-urban refers to areas on the periphery of an urban area, with both formal and informal housing.

\*\* Decade in which the facility was established;

\*\*\* Daily average patient load

selection also accounted for rural and urban locations and whether health clinics were purpose-built or not. Table 2 presents descriptive data on selected clinics.

Working in pairs or small groups, five female researchers with training in public health and qualitative research, as applied to the study of health systems, conducted the fieldwork. We approached the clinic as a "*dynamic site of interaction between humans, microbes, and materials*" [26]. An ethnographic approach was adopted to explore the complex interactions of health facility structures, processes, and behaviours influencing TB-IPC practices within clinics. Over a period of 1 to 3 days in each facility visited, we collected data using unstructured and structured observations, in-depth semi-structured interviews with clinic managers and health workers, and informal conversations with health workers (Table 2). During their time in the clinics, researchers observed their surroundings and took notes on clinic features and processes pertinent to the implementation of TB-IPC. Interviews were conducted with relevant healthcare staff when possible and convenient. Using a guide that was loosely structured around the parameters of recommended TB-IPC measures, we conducted observations in each of the six facilities, in particular focusing on waiting rooms, reception areas and clinical

consultation rooms. Observations in these spaces explored (a) the material differences in spatial layout; (b) workplace design and ventilation; (c) staff and patient movements around spaces and measures taken to encourage or discourage patient congregation (e.g., triaging, fast-tracking); (d) power dynamics and the interactions between staff members, patients, and staff members and amongst patients. Although our time in the facilities was limited, the relatively structured focus on observations allowed the researchers to capture pertinent features of clinic spaces, processes, interactions, and practices that had bearing on TB-IPC through fieldnotes.

We carried out semi-structured interviews with four clinic managers, a nursing service manager, a senior health worker, an IPC manager and two nurses, one specializing in TB. The interviews were aimed at gaining insights into participants' roles and experiences in relation to TB-IPC within each clinic. Questions focused on (a) the nature of services provided to patients, (b) the current IPC plans and protocols regarding patient management, (c) environmental controls, (d) administrative and governance practices, (e) staff roles and responsibilities, (f) respiratory protection practices, (g) enablers and challenges to implementing IPC measures, and (h) individual risk perception and risk management. We held informal conversations around the local implementation of TB-IPC with six healthcare workers, including two TB nurses, a doctor, an IPC manager, an HR manager and a radiographer. Researchers captured the essence of these conversations through fieldnotes. These notes were integrated with the notes from our observations and formed the basis of 'thick descriptions'.

## Data analysis

'Thick descriptions' were created for the facilities visited, based on observations, fieldnotes, and informal conversations that researchers had during the visits to the six clinics, guided by an ethnographic approach [31]. These documents of between 15–20 pages included researchers' hand-drawn maps, photographs of clinic spaces (without any identifying information or individuals in the images), and descriptive as well as reflective notes on the broader working environment, care processes, and IPC practices as observed at the six facilities. All semi-structured interviews were audio-recorded and transcribed verbatim. Data from thick descriptions and interviews were first read through and annotated through open coding by two of the authors (SA and KK) with expertise in global health and medical anthropology. Using a framework analysis approach, a coding framework was developed, informed by a systems-thinking perspective. Specifically, we drew on categories from a health policy and systems research framework [13] distinguishing between health system 'hardware' components including (i) human resources, (ii) organisational structure, (iii) medicines and technology; (iv) service infrastructure, (v) information systems, and (vi) financing and 'software' elements, described as (i) relationships and power, (ii) values and norms, (iii) interest and ideas. We defined what we meant by 'hardware' and 'software' as applied to TB-IPC (Table 3) based on Zwama et al., (2021) and De Bono et al. (2014), and then created a matrix that juxtaposed 'hardware' and 'software' elements shaping TB-IPC processes and practices as observed and recounted by the study participants.

We used a deductive approach to manually code all textual data. A compare-and-contrast method was used to check the consistency of coded content. Displaying the coded segments in the matrix allowed us to interrogate our data for interactions within and across 'hardware' and 'software' dimensions. For the most part, this was a useful way of capturing systemic processes underlying TB-IPC in the clinic environment. However, coding also revealed the often arbitrary division between constituted 'hardware' and 'software'. For instance, one element of systems 'hardware'–governance–compromised both tangible elements such as committees,

**Table 3. Coding framework.**

| Categories | Subcategories | Definitions |
|---|---|---|
| | Human resources | Staff availability, staff ongoing training, staff workload |
| | Organisational structure | Governance, logistics, coordination, support and supervision systems, organisation of care and service delivery |
| Systems Hardware | Medicine and technology | Supply of medication, PPE, air-conditioners/fans, UVC lights |
| | Service infrastructure | Physical infrastructure, spaces, layout and ventilation |
| | Information system | Record systems, information dissemination and communication |
| | Financing | Funding availability |
| | Relationships and power | Climate communication, collegiality, authority and autonomy |
| Systems Software | Values and norms | Institutional support for ICP, accepted practices and normative behaviours in relation to IPC protocols, staff beliefs and perceptions of TB risk |
| | Ideas and interest | Staff professional identify, morale and motivation, job satisfaction, responsibilities and expectations |

guidelines, and protocols for TB-IPC as well as intangible systems 'software'–power relations and social hierarchies that characterised local manifestations of leadership, collegiality and normative practices. In presenting the findings, we eschewed a rigid distinction between 'hardware' and 'software' elements of TB-IPC by highlighting their dynamic interaction within clinic infrastructure, processes, and practice.

## Results

We structure our findings within four categories commonly included in checklists for assessing TB-IPC [8]. In addition to the recommended hierarchy of controls (administrative, environmental, and personal protective), these encompass managerial measures intended to guide the stewardship of IPC activities at a facility level. Although managerial and administrative controls are sometimes combined, we have reported on these separately as they relate to different aspects of organising TB care and TB-IPC (governance versus service delivery). While the control levels are often classified and ranked separately according to their relative effectiveness, they are, in practice, inter-connected. Infrastructural, procedural, and behavioural components of IPC are not implemented on a 'blank slate' but within the living working environment of the clinic. For this reason, we start by describing some of the less visible contextual features of the primary health clinic context that provide an important backdrop for our findings on IPC.

### The primary care context of IPC implementation

"Fixing public health care will require a radical transformation, moving from a system organized at a district level, to a team-based approach focused on patients. Guidelines, of course, must be central to how care is organised, but those guidelines must be contextualised to the existing working conditions [. . .]. Many doctors like myself are deeply anxious about these working conditions. But we are told that we must accept these organizational structures, ways of working, and performance goals expected at a district level." [Doctor, informal conversation, Clinic 3]

The quote above reflects the frustrations that many clinic staff expressed with regard to a mismatch between goals and guidelines set at the district level and their implementation within the working environments of primary health clinics. The doctor refers specifically to problems in the organisation of care and working conditions, two features of clinics that are in

turn linked to the availability of human and material resources, as well as management capacity.

In many clinics, managers were confronted with persistent human resource gaps that affected the delivery of services and uptake of interventions. The funding for filling staff vacancies and procurement of necessary material goods and services followed bureaucratic procedures involving the submission of lengthy forms to affiliated district hospitals and approval of funding by a committee overseeing general clinic spending. Ironically, a clinic manager described how staff shortages contributed to missing the important "cash flow meeting" where the clinic's budget had to be negotiated and approved.

*"We are allowed to be a part of the cash flow committee. The only problem is, there is the hospital cash flow and there is the clinic cash flow. Yeah, so at times you will find that we are supposed to have a clinic cash flow and . . .I'm held up at work due to staff shortage. I cannot attend that cash flow."*

[Clinic manager, interview, Clinic 2]

*"They only gave me budget in March, with a bureaucracy of ratifying posts and everything. Even today, I have not appointed anybody, but I have done everything in terms of interviews and everything."*

[Nursing service manager, interview; Clinic 4]

These challenges were described as longstanding and set within the lack of decentralised power over clinic allocation of funds and spending. Policies and procedures were developed and enforced at the provincial level, with little consideration given to local needs and concerns at the clinic level. A clinic manager lamented the lack of responsiveness of hospital management towards requests articulated for specific basic equipment.

*"It's rare where they* [management at the hospital] *accept our requests because of the functional cost saving that we have to make as a Department. Like we need a backup generator. We don't have a backup generator, and they* [at the hospital] *say it's too expensive for now.*

[Clinic manager, interview, Clinic 6].

The organisation of care in this setting was also seen to be subject to the vagaries of donor funding, changes in disease priorities and in health policy. For example, the manager of Clinic 4 described how the withdrawal of funding for HIV through an external NGO led to a loss of critical administrative staff including data entry clerks. This had a knock-on effect on filing systems and patient flow. In another clinic, the manager described the challenges of meeting the targets of the ICI, which had unintended negative consequences for both the workflow and patient flow.

Front-line staff talked about tensions in the workplace, leading to high staff turnover and understaffing, and further impacting job satisfaction. For example, many health workers referred to unmanageable daily workloads leading to fatigue and the feeling that they were overlooked by management.

*"I feel exhausted, tired and sleepy, because to nurse more than 100 patients per day, sometimes you become very exhausted . . .. Other people think you need to walk around to do heavy work, then you become tired. But to sit down and talk, it's very exhausting. I enjoy talking to patients* [. . .] *but if it's 100, or 70, or 80* [patients], *sometimes it's too much.*

[Nurse, interview; Clinic 2]

Staff expressed feeling excluded from participation in decision-making on matters related to their daily practice. The management style was described as "top-down', contributing to the feeling of powerlessness. In Clinic 4, one health worker commented:

*"I am tired of a "dead-end job", where I can make no difference. There is no hope. I want to be in a job where I can see that what I do makes a difference. Where I can reflect on myself and modify my actions to make the outcomes different".*

[Health worker, informal conversation; Clinic 4]

However, staff in Clinic 6 talked at length about the new clinic manager perceived as a "*breath of fresh air*" in the way things were run in the clinic, including some aspects of normative organisational behaviour around IPC. They described the management style as inclusive, responsive and attuned to clinic needs. Staff were often praised for their good performance and professional commitment. This had a positive effect on staff general morale and practice, which, in turn, helped to attain *"the Ideal clinic gold status"*.

*"Our clinic manager is responsive to staff concerns and requests. He is inclusive in decision-making and implementing changes after discussions with us. He asks for solutions from us as a team, considers our perspective and shares information with us".*

[Health worker, informal conversation; Clinic 6]

## Managerial measures

Managerial functions underpinning stewardship of TB-IPC are most visibly embodied in the presence of infection control plans, dedicated focal persons and committees for TB-IPC. In all clinics visited, managers affirmed that IPC protocols, which TB-IPC measures are part of, were easily accessible to all health workers. In some clinics, IPC committees were formed, or IPC champions were assigned the responsibility of monitoring the implementation of IPC protocols. However, there was ambivalence about the effectiveness of IPC committees:

*"They have committees on paper, because I don't remember. . ...I think it was in 2018, early, when we had a meeting with them* [IPC committee]. *We do have them on paper. When the auditors ask—Who are the committees for IPC, I will pull the file and show you, because we have appointed people. But in terms of minutes and everything no, it's not that active like they should be."*

[Health worker, interview; Clinic 4]

Similarly, the effectiveness of designated IPC champions was not always evident. For example, one of the IPC champions we met was unaware of the checklist of TB-IPC measures they were tasked with monitoring. In Clinic 3, the IPC manager mentioned that TB-IPC guidelines were an integral part of the general IPC file, but that these had last been reviewed in 2012. While this manager was aware that routine monitoring of TB-IPC measures was meant to occur every six months, the most recent TB-IPC checklist had been completed 18 months prior to our visit.

Clinic managers highlighted the lack of earmarked funding for IPC activities—*"no, in our financial expenditures there is no specific budget for IPC"* [Clinic manager interview, facility 6]–which impacted the availability and frequency of staff training. In one clinic, a nurse mentioned that staff training was contingent on provincial funding:

*"We are waiting for the budget, because we are closing the previous year now at the end of this month* [March 2019]. *So, we are hoping that they* [Department of Health] *will appoint the service provider to come and assist in terms of training and programming system."*

[Nurse, interview; Clinic 4]

In Clinic 6, the TB nurse reported that she had last received training on TB eight years ago. She recalled that the training had "*covered the basics*" including personal protective equipment (PPE) and that she was subsequently able to offer in-service training on the use of N95 masks. However, as shown in a later section, the sustained impact of this training is not evident. IPC is an area that is neglected, yet an easy target for unwelcome audit exercises:

*"There is an IPC team at the district hospital. They told her about training 2 years ago and she still hasn't heard from them until today. When the minister of health comes for a visit, that's when they* [district hospital IPC team] *come the day before to stand behind you and ask all the questions about things that they have never told you about".*

[TB nurse, interview; Clinic 2].

Screening and follow-up investigations for TB in staff also fall under the remit of clinic management. However, we observed uneven attention to occupational health and the safety of front-line staff. An IPC manager from one of the clinics admitted they felt short of facility-based surveillance of HCAI, including periodic screening of staff for active TB.

*"TB screening is supposed to be done annually for all staff, and for some staff at high risk, it should be done every six months. Staff are often busy and delay coming for their assessment or re-schedule."*

[IPC manager, interview, Clinic 4]

Conversely, the manager in Clinic 6 reported that they had initiated regular TB screening of staff. This measure was intended to improve on past practice, when only new staff, or those working in secondary hospitals and not primary care, were screened for TB. TB notification among the staff was, however, noted to be a rare event.

*". . ..we do it [screening] here around April, where we take sputum for all the staff and we do GXP [GeneXpert]. If the results come back positive, we refer [the staff] to the Hospital. But since I came here in September 2015, no one has been diagnosed with TB among the staff members".*

[Clinic manager, interview, Clinic 6]

## Administrative controls

Administrative measures for TB IPC are largely concerned with the logistics of organising patient flow and services in ways that reduce the risk of exposure to undiagnosed TB. In all except one clinic, health workers talked about how acute staff shortages and frequent staff rotations affected timely case identification and triaging of patients for undiagnosed TB. In Clinic 2, the registering nurses were seeing two streams of patients at the same time in the same room due to staff shortages. In Clinic 5, the triage nurse was pulled in temporarily for processing the results of blood tests. In Clinic 3, the manager described triaging as a "*grey area*" not only because of staff non-availability but because of restrictions on who was permitted to triage:

*"There is no one who is able to pitch in and say—'I will be able [to triage].' We have tried all doors, but the staff availability we don't have. We've tried with Mr [name of Security Guard]. But he is not clinical, and we cannot use him. But with emergencies they are trying."*

[Clinic manager, interview; Clinic 3]

Waiting areas in most clinics were often overcrowded, particularly in the morning hours early in the week, when queues started to build up from four in the morning. Commonly, there was limited signage to guide patients through clinics, contributing to patients' congregating in confined spaces and thus increasing the risk of TB transmission. Most clinics did not have functional appointment systems or were only starting to introduce them.

*"Patients come any time. But now, what we are encouraging is the booking system because we have seen that we have got floods and floods of clients, and there is no control over who is coming and from where. . . . . . .there may be the abuse of services and the booking system may bring that accountability".*

[Clinic manager, interview; Clinic 3]

Clinic managers referred to cultural norms around health-seeking behaviour that were unconducive for setting up formal appointment systems. There was also awareness from service providers that patients relying on public sector clinics are also likely to be reliant on public transport, which in many instances is limited and does not adhere to reliable schedules. This can make it challenging to stick to rigid appointment times and requires services to adapt to contextual realities. Patients were accustomed to waiting many hours and, therefore, did not stick to appointment times out of fear of missing their turn. As a result, as reported by one clinic manager, "*patients are given return dates and not times. Patients need to become accustomed to returning on the appointed date before getting them accustomed to an appointment time*". [Clinic manager, interview; Clinic 1]

In some cases, the adoption of a streamlined centralised filing system in response to an ICI policy directive had negative consequences for the patient flow:

*"That's why we see these long queues. It's the influx of patients. [. . .] We did not plan for that. So even with the appointment system we did not step their appointments. They were just given the same date. So, we had an influx of patients that were coming in on one day, more than 700 to 800 a day. It was very difficult. The waiting time was long. Some patients would come at 5:00am. They will be sitting there with no cards until 16:00pm, and then would be told that: 'We can't find your file. You have to open a new file.'*

[Nurse, interview; Clinic 4].

A number of clinics relied on the manual retrieval of folders and manual recording of patient data further contributing to waiting times. In Clinic 4, equipment for electronic recording was available; however, the gaps in logistics and training stalled their integration into care practice.

*"We have got the equipment* [59 computers], *but our computers are not programmed, and staff is not trained, so we are waiting now for the Department* [of Health] *to send the trainers . . . At the moment everything is just manual paperwork.*

[Health worker, interview; Clinic 4]

The implementation of administrative measures for TB-IPC involves managing the organisation of care effectively, which in turn is closely interlinked with human resources, information systems and organisational behaviour practices.

## Environmental controls

Measures to improve the environment for prevention of *Mtb* transmission generally focus on the clinic space and material infrastructure, yet the use of space, and the movement of people and air within clinic spaces are influenced by the organisational and behavioural norms. Staff across all the clinics were well aware of crowded waiting areas and poorly ventilated clinic spaces as *"hot spots"* for transmission of *Mtb*. However, in some cases, inconsistent use of clinic spaces designated for particular services increased the risk of *MTb* transmission. For example, in one clinic, the labour ward was interchangeably used for patient consultations; in another, a room intended for TB care was being used for integrated management of childhood illnesses. In some clinics, patients did not have appropriate spaces to produce sputum samples, challenging basic tenets of biosafety:

> *""If you see there—a room we are using currently for children >5. . ..but the way it was built it was meant for TB, as there is an area at the back for coughing. If I tell the patient to go and cough outside, he will go anywhere. So that are some of the gaps I have identified that we need to have sort of an area* [for coughing]. . ."*

[Clinic manager, interview; Clinic 6].

> *"I think this facility needs a well-ventilated coughing booth, instead of allowing patients to cough outside behind the bathrooms. That is the designated area for now, but you will find patients producing sputum anywhere they see fit. Now, I prefer the patient to produce the sputum at home"*

[Senior health worker, informal conversation, Clinic 3].

In Clinic 3, one doctor described how he was lobbying to have a TB testing laboratory and equipment to speed up the reporting of individuals' test results. Current arrangements meant that the patient had to return for test results after 48 hours, contributing to high rates of loss-to-follow-up and the failure to initiate individuals on treatment in a timely manner.

Generally, however, many clinic staff expressed being powerless to influence the allocation of space for clinic services. Decisions regarding the use of clinic space and procurement of equipment fell under district-level management:

> *I requested for the park home whereby they are going to separate TB patients; and they* [management at the hospital] *found it as a financial burden and said*: *"Let's try our partners* [NGO]. *Maybe they would provide you with the park home." Our supportive partner is* [name of NGO]; *but until now—nothing!"*

[Clinic manager, interview, Clinic 6].

A key environmental control for TB-IPC is ventilation. We observed that few clinics kept windows and doors open, limiting indoor air circulation. Ensuring adequate natural and mechanical ventilation often conflicted with perceived levels of personal comfort. For instance, most clinics were equipped with fans and air-conditioners which were used on very hot and humid days, while windows and doors were kept closed in confined spaces. The use of

mechanical ventilation was often reduced to a minimum to comply with cost-saving requests imposed by leadership.

In other clinics, the equipment for mechanical ventilation was not functional. A sense of apathy regarding the possibility of getting equipment repaired was pervasive:

> *"The fans are not working, and the ventilators are not functional.. . ..the maintenance is provided by the guy from district. . ..Yes, reported* [equipment is dysfunctional]. *Ummmmm. Eish. . . I don't remember* [when last reported].

[Health worker, interview; Clinic 5]

Ultraviolet (UV) lights were absent in the majority of observed clinics. In Clinic 3, the IPC manager noted that UV lights are: "*. . .expensive to purchase and the maintenance is an issue. The DOH* [Department of Health] *doesn't want to invest in them*". More generally, slow and bureaucratic procurement tender systems led to delays and in some cases, provision of substandard items:

> "*You want something of quality, but they are always only accepting the cheaper solution. For instance, there was a time that we used gloves that broke easily and did not fit properly. This led to an incident.*"

[IPC manager, informal conversation, Clinic 3]

## Personal respiratory protection

We observed variable responses to the use of personal respiratory protection in clinic spaces. We note two interacting themes–the low perceived risk of TB acquisition and organisational normative practices around PPE that contributed to poor compliance with personal respiratory protection measures. PPE including N95 particulate respirators and surgical masks were in sufficient supply across all clinics, however, these were not consistently in use when required.

> *"We have masks, but you know, nobody wants to wear a mask the whole day sitting there. But we do have in all the service areas. We have got the N95 and the surgical masks. We are supposed to be using the N95 masks in all the service areas and then the surgical mask whenever dealing with the patient".*

[Clinic manager, interview; Clinic 1]

> *"It's a challenge to wear a mask, because other HCWs may think that you are making them look bad, and so do patients. If it is a uniform measure, it would have been easier."*

[Speech therapist, informal conversation; Clinic 2]

In one clinic, an occupational health nurse reported periodic assessments to evaluate the acquisition risk for TB and other infections. Assessments included checks on staff's wearing of N95 respirators; non-compliance was recorded in books in case staff developed TB, implying this was largely viewed as a matter of individual choice and responsibility.

We noted that respiratory protection programmes were not effectively operationalised and training of staff on respiratory protection was sporadic. One staff member recalled undergoing no practical training on donning or removing PPE. Another staff member had misperceptions about the types of face-coverings:

*"The N95 mask protects the nurse from getting an infection from the patient but does not protect the patient from getting infected by the nurse."*

[Nurse, interview; Clinic 1]

The same staff member also reported disposing of N95 masks after each use as soon as practically possible, suggesting a potential lack of awareness on proper usage, disposal and reuse of respiratory protective devices.

We probed staff on the perceived risk of contracting TB; staff expressed the (fallacious) opinion that hospital-acquired TB was a rare event or that the risk of TB was located in specific spaces or patients (e.g., patients taking MDR-TB treatment, which was not delivered in these clinics). A TB nurse reported that mask-wearing was not a working norm or practice but guided by staff members' *"own judgement of risk"* as well as their threshold of discomfort in wearing a mask:

*"I have never contracted any infection. Actually, we are supposed to wear masks. But . . .I personally fail to wear a mask because it creates a mist on my glasses, then I cannot see. If it happens that I touch the patient, I do wash my hands. Even before. There is a spray."*

[Senior health worker, interview; Clinic 5]

Mask-wearing, therefore, was not seen as an institutional norm and enacted collective behaviour but rather, an individual choice based on the perceived risk of acquiring infection. Linked to the lack of a distinct ethos of patient and health worker safety with regards to the transmission of TB, it is perhaps not surprising that staff developed alternative narratives of personal protection. When asked how they protected themselves, practices such as *"keeping a good distance from patients"*, *"having a good healthy breakfast"* and *"asking for protection from God"* indicate that for some health workers, the risk of contracting TB might be inherent to working in this setting but not perceived as an imminent personal threat requiring concerted action.

## Discussion

Using a clinic ethnography approach, we highlight the interplay of 'hardware' and 'software' elements shaping TB-IPC within six primary healthcare clinics in KwaZulu-Natal, South Africa. By examining gaps between evidence-based TB-IPC protocols and observed practices within the existing real-world management structures, the organisation of care and the working culture of clinics, we expand current understandings of the 'enabling environment' required for successful uptake of IPC interventions.

IPC measures imply changes to the way facilities and staff within them work—policies have to be translated and taken up at the clinic level, services may need to be re-organised, and staff relativise the value of IPC in relation to other personal and professional priorities. Consequently, a better understanding of the context of implementation has implications not only for how to improve monitoring of IPC practice but also for how to embed an ethos of IPC within clinics.

In the primary care clinics included in this study, managers and front-line staff negotiate and adapt work practices within the constraints of existing material infrastructure and available human and financial resources. We found that wider governance issues affected funding and staffing levels, and, in turn, the availability of a dedicated and trained team for overseeing IPC. The impact of these constraints was visible when examining how specific IPC measures were operationalised (or not). Within each of the four dimensions conventionally assessed in

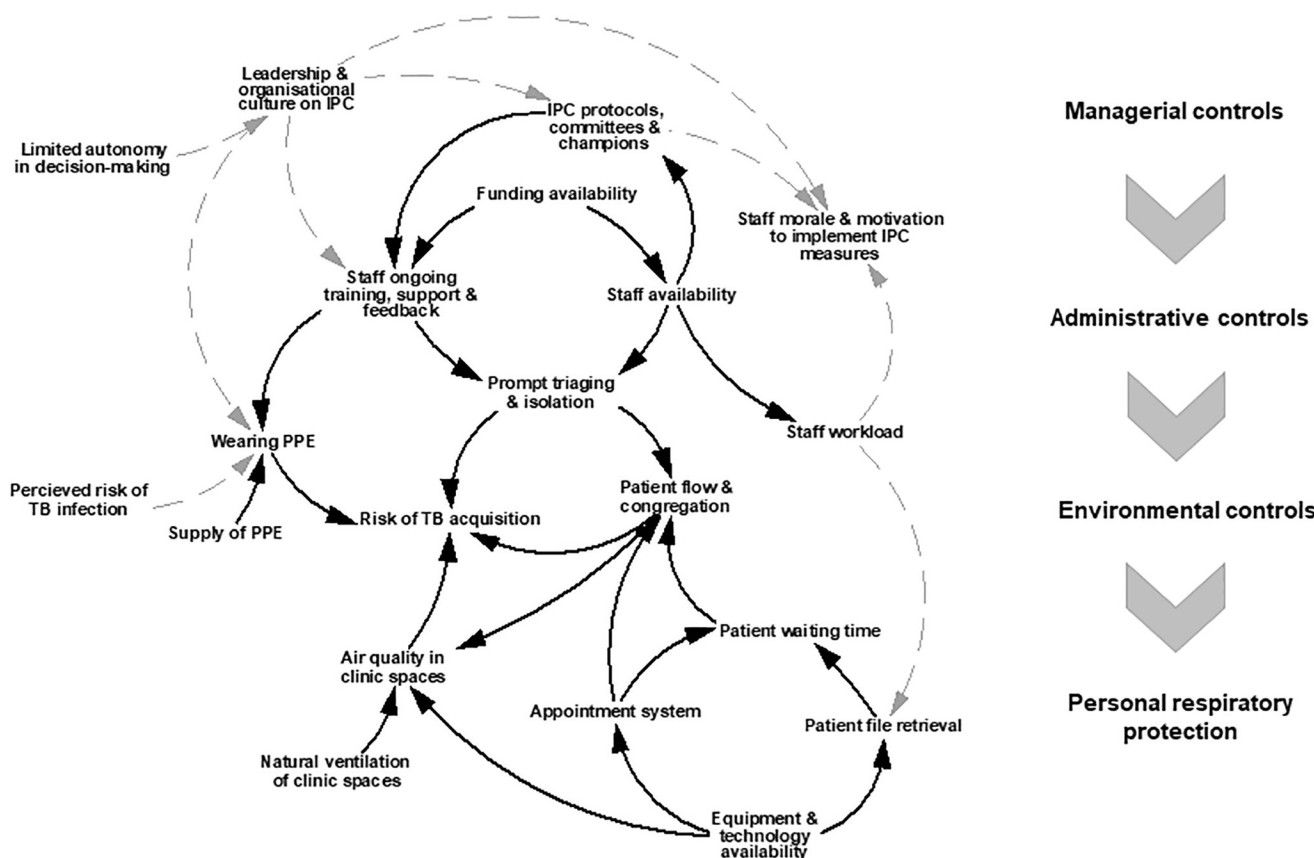

**Fig 1. Key interactions between health systems hardware and software components across the hierarchy of TB-IPC controls.** Black arrows show tangible interactions between health system hardware components. Dashed grey arrows show less tangible interactions between health systems software and hardware components.

TB-IPC checklists, we noted interactions across 'hardware' and 'software' components. The key interactions are illustrated in Fig 1.

The visible 'hardware' of *managerial* measures, that is, IPC protocols, IPC committees and IPC champions were available in principle, but their utility and functionality in the prevention of HCAI were not obvious to staff; this was linked to inadequate or limited staff training. From staff descriptions, a suboptimal application of managerial and administrative controls was associated with resource constraints, and the lack of support from the district-level management, underscoring the heavy reliance on external support for addressing internal deficiencies.

Developing, executing and evaluating IPC improvement strategies which are embedded within a cycle of clinics' IPC and quality improvement strategies is essential. Clinic managers and senior administrators have a central role to play in this by acting as role models and training and supporting staff in building capacity and proficiency in the procedures necessary for working safely [32]. Stand-alone staff training is unlikely to achieve expected outcomes unless staff is contiguously primed on how to navigate opportunities and barriers in applying their knowledge. Low hanging fruit for improving IPC such as outdoor waiting areas can be achieved with limited staff effort and does not rely on radical behavioural change.

Challenges in implementing *administrative* measures including triaging and screening, for example, were linked to the inefficient organisation of patient flow and information, in turn, a consequence of norms in the organisation of care, staff shortages, and in some cases

unintended consequences of top-down policy directives. Suboptimal implementation of TB-IPC in the South African context resulting, at least in part, from the conflicting guidance brought by the new policy initiatives has been described previously [24]. This emphasises the importance of aligning guidance issued for IPC with other health policy initiatives.

We further note that the integration of *environmental* controls in the working life of staff in facilities was challenged by the interplay of infrastructural and organizational issues. More specifically, limitations in the material and physical infrastructure interacted with clinics' lack of financial and decision-making autonomy to limit the potential of relatively simple solutions for reducing transmission of *Mtb* in clinics. While the success of IPC is often predicated as contingent on 'hardware' known effective interventions, simple measures such as diluting and removing contaminated air by ensuring natural ventilation are neither costly nor require sophisticated equipment.

*Personal protective* measures, though familiar and available to clinic staff, were also viewed with ambivalence, in part due to a lack of perceived risk, but also the lack of a collective ethos around health worker and patient safety. It is essential to note that one clinic stood out in terms of reflecting the facilitative effect of positive organisational culture and climate on staff morale and adherence to IPC measures. In this clinic, staff responses to IPC were influenced by participatory support from senior management and working relationships that encouraged proactive ownership and facilitation of quality improvement initiatives in IPC, eliciting effective and sustained change.

Developing shared assumptions and beliefs around good IPC practice–a 'collective mental shift'–and translating these beliefs into care practice should be a key consideration. This reinforces the need for focusing on the social qualities of organisational culture and developing people-focused strategies that would motivate, empower and encourage staff involvement in clinic-wide IPC improvement processes while responding to identified barriers and needs. The central role of organisational culture in facilitating good IPC practices has been shown in previous research [3, 5, 10].

Recognising that the hardware and software of TB-IPC are mutually constituted also suggests that we must go beyond the current checklist approach to assess the implementation of practice and factors that influence the ability to take action. Clinic ethnography methods allowed for multiple entry points and voices to the 'problematic' of compromised TB-IPC, highlighting that it is important to capture dimensions of the so-called 'enabling environment' that are currently not assessed. These include dimensions of management and organisational culture, as well as features of the policy and systems initiatives that are in place, and that influence the practical and logistic implementation of administrative and environmental measures, in particular. We suggest that tools to monitor not only TB-IPC practices, but IPC more generally, should be more carefully designed to promote their use as participatory and supportive forms of supervision rather than binary audits that do little to promote either understanding of the reasons behind 'poor performance' or encourage behavioural and organisational change. Enhanced checklists can incorporate short sections that provide more detail on relevant aspects of clinic governance, funding, and infrastructure as well as questions that elicit more granular information regarding *how* IPC measures are functioning, and open-ended responses to elicit systemic challenges (or conversely changes) that have hindered or facilitated 'good' IPC practice.

## Limitations

Our work focused on the implementation of TB-IPC in six clinics and in one province; the findings may not be generalizable to other provinces of South Africa. However, this is inherent

to our method: ethnography does not aim to achieve generalisability, but rather an in-depth analysis of the meaning of practices and interactions in context. Our findings are transferable in that they indicate what dynamics must be considered when studying the implementation of TB-IPC, or indeed IPC practice more generally. At the same time, we recognise the complex dynamic that plays itself out in individual clinic settings is unique, and the balance of forces in each setting is differently nuanced, which means interventions to modify 'software' issues discussed in this paper need careful co-adaptation.

We had limited contextual data collected through direct observations from Clinic 5. We, however, believe that in-depth accounts from a senior health worker provided rich and detailed descriptions of TB-IPC practices in this clinic. Interpretation of our findings was influenced by researchers' individual perspectives, inherent to an ethnographic approach, yet at the same time balanced through consultation and discussion with the wider multi-disciplinary team that we are part of.

Finally, our fieldwork was undertaken before the global outbreak of COVID-19, and some IPC practices in clinics, for example, mask-wearing, are likely to have changed as a result of new directives. Nonetheless, our insights on the wider dynamics of systems hardware and software remain important for sustaining and improving TB-IPC practices in resource-constrained TB-burdened settings.

## Conclusions

Optimal implementation of TB-IPC requires strategies that contextualise TB-IPC processes and practices within the structure and functioning of the whole system. There is a pressing need to adapt the implementation of multimodal measures recommended by WHO (2019) to local structural and operational constraints of primary health clinics, as well as considerations of security and comfort for staff and individuals attending clinics. Addressing these issues requires ensuring there are enough human and material resources to implement recommended IPC measures. However, while resource constraints influence optimal IPC implementation, the observed variation in IPC practices under similar conditions points to the centrality of human factors. Addressing structural and staff shortages might not be sufficient to enhance IPC practices unless organisational management, clinic-wide participatory planning and coordination of strategies around IPC are functioning well. Health workers' performance is dependent on their motivation to confront challenges and act as agents of change. Person-centred approaches that would motivate, empower and encourage staff's integration of IPC improvement processes into care practices while responding to identified barriers and needs should be considered. Using participatory tools to elicit the views of clinic implementors and going beyond audit-style checklists can allow not only identification but also space for reflection on the obstacles and leverage points for change.

## Supporting information

**S1 Checklist. Inclusivity in global research.**
(DOCX)

## Acknowledgments

We acknowledge the important contributions of Gimenne Zwama, Thandeka Smith and Zama Khanyile in conducting and documenting the ethnographic fieldwork in KZN. We are grateful to the health clinic managers and health workers for their time, interest, and

motivation to participate in this study. We thank our colleagues from the *Umoya omuhle* project for the rich inter-disciplinary discussions and exchange of ideas.

## Author Contributions

**Conceptualization:** Stella Arakelyan, Hayley MacGregor, Anna S. Voce, Janet Seeley, Karina Kielmann.

**Data curation:** Hayley MacGregor, Anna S. Voce, Karina Kielmann.

**Formal analysis:** Stella Arakelyan, Karina Kielmann.

**Funding acquisition:** Anna S. Voce, Alison D. Grant, Karina Kielmann.

**Investigation:** Anna S. Voce, Karina Kielmann.

**Methodology:** Hayley MacGregor, Anna S. Voce, Janet Seeley, Karina Kielmann.

**Project administration:** Alison D. Grant.

**Supervision:** Janet Seeley, Alison D. Grant, Karina Kielmann.

**Writing – original draft:** Stella Arakelyan, Karina Kielmann.

**Writing – review & editing:** Stella Arakelyan, Hayley MacGregor, Anna S. Voce, Janet Seeley, Alison D. Grant, Karina Kielmann.

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
