## [Decision Letter · Decision Letter 0]

21 Jun 2022

PGPH-D-22-00745

Beyond checklists: using clinic ethnography to assess the enabling environment for tuberculosis infection prevention control in South Africa

Dear Dr. Kielmann,

Thank you for submitting your manuscript to PLOS Global Public Health. After careful consideration, we feel that it has merit but does not fully meet PLOS Global Public Health’s publication criteria as it currently stands. Therefore, we invite you to submit a revised version of the manuscript that addresses the points raised during the review process.

Please submit your revised manuscript by . If you will need more time than this to complete your revisions, please reply to this message or contact the journal office at globalpubhealth@plos.org. Please include the following items when submitting your revised manuscript:

We look forward to receiving your revised manuscript.

Kind regards,

Melanie Boeckmann

Academic Editor

Journal Requirements:

State what role the funders took in the study. If the funders had no role in your study, please state: “The funders had no role in study design, data collection and analysis, decision to publish, or preparation of the manuscript.”

3. Please ensure that the funders and grant numbers match between the Financial Disclosure field and the Funding Information tab in your submission form. Note that the funders must be provided in the same order in both places as well.

4. Please update your Competing Interests statement. If you have no competing interests to declare, please state: “The authors have declared that no competing interests exist.”

Additional Editor Comments (if provided):

Dear colleagues

two reviewers have independently assessed your manuscript and recommend minor revisions. I would encourage you to address their comments and re-submit a revised versison.

Reviewers' comments:

Reviewer's Responses to Questions

**Comments to the Author**

1. Does this manuscript meet PLOS Global Public Health’s publication criteria? Is the manuscript technically sound, and do the data support the conclusions? The manuscript must describe methodologically and ethically rigorous research with conclusions that are appropriately drawn based on the data presented.

Reviewer #1: Yes

Reviewer #2: Yes

2. Has the statistical analysis been performed appropriately and rigorously?

Reviewer #1: N/A

Reviewer #2: N/A

3. Have the authors made all data underlying the findings in their manuscript fully available (please refer to the Data Availability Statement at the start of the manuscript PDF file)?

Reviewer #1: Yes

Reviewer #2: Yes

4. Is the manuscript presented in an intelligible fashion and written in standard English?

Reviewer #1: Yes

Reviewer #2: Yes

5. Review Comments to the Author

Reviewer #1: The manuscript is an exceptionally well written piece of qualitative research that represents the culmination of ethnographic observation, interviews and informal conversations on TB infection prevention and control (IPC) in primary health clinics in a high-TB-burden province of South Africa. The researchers seek to detail IPC practices in a well-conceived purposive sample of 6 clinics within the province. The data, explored through a systems approach, exposes less tangible measures, or what the researchers refer to as ‘software’ issues (“relationships and power, values and norms, ideas and interest”) that contribute to the challenges of implementing IPC, in addition to the usual ‘hardware’ issues (e.g. human resources, infrastructure, and PPE) that are most often considered. Grounded within the data reported, the researchers discuss ways in which some of these software issues could be better considered through organisational leadership and greater input from clinic staff. This manuscript contributes important lessons to help strengthen TB IPC at a time when COVID regulations - which resulted in an upsurge in concerns about health worker safety and the strengthening of IPC measures – are on the cusp of being withdrawn, and South Africa’s National TB Programme is pushing to sustain and improve IPC policy and implementation through its upcoming strategic plan.

Minor comments for clarity are as follows:

Methods:

Page 8, Table 2:

- It would be helpful to the reader to include a brief description of how rural, semi-rural, peri-urban, and urban were defined in the main text for the purposes of clinic selection.

- It is not clear how the governance structures of "KZN government" and "KZN Provincial government" clinics differ. Were only provincial clinics selected due to convenience?

- The observation periods are quite short for ethnography. Typically ethnographic observation extends over a period of weeks to reduce the Hawthorne effect, yet, the data do not suggest being observed resulted in a change in behavioural patterns. It would be worth adding a few sentences under methods about how the observation was carried out to provide readers with a bit more context.

Page 9, line 183: I found it a bit confusing when the authors referred to the researchers as “participating” in the fieldwork, as “participants” are often those who are the subject of the research. Consider replacing with “conducted” or “undertook.”

Page 9, line 202: what is meant by “service users”? Is this strictly patients, or might healthcare workers be considered service users in this context?

Page 9, 2nd paragraph:

- Did interviews happen before or after observation?

- How were informal conversations documented?

Ethical considerations (Lines 246-263):

It would be helpful to include a few lines about how the use of data from informal conversations would be covered under institutional approvals given the nature of ethnographic work.

Results

Line 324: “In other clinic…” should read “In another clinic…”

Lines 368-371: These two sentences were a bit confusing to read. Consider re-wording to make the point a bit clearer.

Line 399: What is meant by staff working “at secondary level”? Does this mean outside primary care?

Line 412: I was unclear on what the authors meant by “registering nurses seeing two streams for patient care at the same time”. Does this mean one nurse had two sets of patients that they would move between, or two nurses each seeing patients at the same time in the same room?

Line 482: “patient TB test results” should read “TB patient test results.”

Lines 543-545: It may flow better to stop the sentence leading into the quotation on lines 547-48 at “Another staff member had misperceptions about types of face-coverings:” and then to add the rest of the sentence after the quotation with some additional context, e.g. “The same staff member also reported disposing N95s after each use as soon as practically possible, suggesting that…”

Discussion

Given the nature of the enquiry, the authors may want to consider saying a little more about a few TB IPC “recommended practices” that may not be ideal or may require more nuance for the setting, e.g.:

- Line2 424-426 includes data on introducing appointment dates rather than times. The concept of patient appointments is often recommended by the privileged without the understanding that a vast majority of public sector patients rely on public transport (ie an informal taxi network), which does not run according to a schedule, making appointment times near to impossible to keep, especially for those living outside of busy transport hubs.

- Line 496 refers to health workers preferring personal comfort, in this case using air conditioning units to keep out heat. This can also be extended to keeping windows open or asking sick patients to wait in outside queues during winter months, which may also bring a level of discomfort and lack of feeling safe for patients or staff that is also overlooked by those who advise on IPC.

Might the global community need to rethink some of these easy to say IPC practices, but not so easy to endure for those who are on the receiving end?

Limitations:

It would be helpful to remind readers that the data were collected pre-COVID, and thus practices may have improved as a result of the COVID-19 pandemic; nonetheless the lessons remain important for sustaining and improving IPC impetus.

It would be worth considering how presenting the purpose of the work to all clinic staff (as indicated under methods) might have affected (or not) the observed behaviour. May the timing of presentation and observation of mitigated this? – or perhaps it was a testament to how normalized it may be not to follow IPC protocols if behaviour remained unchanged despite the purpose of the observation being known.

Reviewer #2: The authors use a clinic ethnographic approach to highlight important considerations for infection prevention and control (IPC) in the context of TB control. Findings on the managerial, administrative, environmental and personal protection measures and their operationalisation and challenges observed, provide potentially useful information for stakeholders in this context and in other high burden TB countries, even as healthcare workers globally battle with COVID-19 control, another airborne infection.

However, a few issues need to be addressed:

• As is sometimes the case with qualitative/ethnographic findings, the thickness of descriptions could get in the way of finding key information quickly. I would suggest a table or an illustrative diagram to summarize the results, with a representative quote where possible.

• An additional level of synthesis of results could also be helpful. One suggestion would be to merge the managerial and administrative control findings together.

6. PLOS authors have the option to publish the peer review history of their article (what does this mean?). If published, this will include your full peer review and any attached files.

**Do you want your identity to be public for this peer review?** For information about this choice, including consent withdrawal, please see our Privacy Policy.

Reviewer #1: **Yes: **Jody Boffa

Reviewer #2: No

---

## [Decision Letter · Decision Letter 1]

12 Oct 2022

Beyond checklists: using clinic ethnography to assess the enabling environment for tuberculosis infection prevention control in South Africa

PGPH-D-22-00745R1

Dear Professor Kielmann,

We are pleased to inform you that your manuscript 'Beyond checklists: using clinic ethnography to assess the enabling environment for tuberculosis infection prevention control in South Africa' has been provisionally accepted for publication in PLOS Global Public Health.

Best regards,

Melanie Boeckmann

Academic Editor

Reviewer Comments (if any, and for reference):

Reviewer's Responses to Questions

**Comments to the Author**

1. If the authors have adequately addressed your comments raised in a previous round of review and you feel that this manuscript is now acceptable for publication, you may indicate that here to bypass the “Comments to the Author” section, enter your conflict of interest statement in the “Confidential to Editor” section, and submit your "Accept" recommendation.

Reviewer #2: All comments have been addressed

2. Does this manuscript meet PLOS Global Public Health’s publication criteria? Is the manuscript technically sound, and do the data support the conclusions? The manuscript must describe methodologically and ethically rigorous research with conclusions that are appropriately drawn based on the data presented.

Reviewer #2: Yes

3. Has the statistical analysis been performed appropriately and rigorously?

Reviewer #2: N/A

4. Have the authors made all data underlying the findings in their manuscript fully available (please refer to the Data Availability Statement at the start of the manuscript PDF file)?

Reviewer #2: Yes

5. Is the manuscript presented in an intelligible fashion and written in standard English?

Reviewer #2: Yes

6. Review Comments to the Author

Reviewer #2: (No Response)

7. PLOS authors have the option to publish the peer review history of their article (what does this mean?). If published, this will include your full peer review and any attached files.

**Do you want your identity to be public for this peer review?** For information about this choice, including consent withdrawal, please see our Privacy Policy.

Reviewer #2: No
